# Detection and Analysis of *Clostridioides difficile* Spores in a Hospital Environment

**DOI:** 10.3390/ijerph192315670

**Published:** 2022-11-25

**Authors:** Zofia Maria Kiersnowska, Ewelina Lemiech-Mirowska, Michał Michałkiewicz, Aleksandra Sierocka, Michał Marczak

**Affiliations:** 1Department of Management and Logistics in Healthcare, Medical University of Lodz, 90-419 Lodz, Poland; 2Institute of Environmental Engineering and Building Installations, Faculty of Environmental Engineering and Energy, Poznan University of Technology, 60-965 Poznan, Poland

**Keywords:** patient safety, risk management, *Clostridioides difficile* spores, C diff Banana Broth

## Abstract

*Clostridioides difficile*, due to its long survival time in a hospital environment, is considered to be one of the most frequent factors in healthcare-associated infections. Patient care requires not only rapid and accurate diagnosis, but also knowledge of individual risk factors for infections, e.g., with *C. difficile*, in various clinical conditions. The goal of this study was to analyse the degree of contamination of a hospital environment with *C. difficile* spores. Culturing was performed using C diff Banana Broth^TM^ medium, which enables germination of the spores of these bacteria. Samples were collected from inanimate objects within a hospital environment in a specialist hospital in Poland. The results of the study demonstrated the presence of 18 positive samples of *Clostridioides* spp. (15.4%). Of these, *C. difficile* spores were detected in six samples, *Clostridioides perfringens* in eight samples, *Clostridioides sporogenes* in two samples and *Clostridioides innocuum* and *Clostridioides baratii* in one sample each. Among the six samples of *C. difficile*, a total of four strains which produce the B toxin were cultured. The binary toxin related to ribotype 027 was not detected in our study. Nosocomial infection risk management is a significant problem, mainly concerning the issues of hygiene maintenance, cleaning policy and quality control, and awareness of infection risk.

## 1. Introduction

The hospital environment plays a significant role in nosocomial infections, since it constitutes a reservoir of pathogens. Many micro-organisms, including *Clostridoides difficile* (*C. difficile*) spores, have the capacity to survive for long periods of time on hospital surfaces, from where they can be easily transferred to personnel or to patients via direct or indirect contact. The risk of infection with this pathogen increases when the patient is present in an environment or room where previously infected or colonised patients or medical personnel were present [1,2,3,4].

*C. difficile* is a Gram-positive, anaerobic spore-forming bacillus widely present in the environment. It produces toxins A (TcdA) and B (TcdB), coded by genes (*tcdA* and *tcdB*) located in the pathogenicity locus PaLoc. These toxins cause, among others things, destruction of the cytoskeleton, apoptosis of the epithelial cells and the induction of proinflammatory cytokine production. Since the beginning of the 21st century, in many countries around the world, not only has the frequency of CDI been on the rise (*C. difficile* infection), but so has its severity. This is related to the appearance of a new, virulent NAP1 (North American Pulsed Field Type 1) strain, also called the PCR 027 ribotype (BI/NAP1/027). This hyper-virulent *C. difficile* strain has the ability to produce a significantly increased amount of toxin A (16-fold) and toxin B (23-fold) and to produce a binary toxin (ADP-ribosyltransferase) which is not produced by other strains. This endemic strain causes a more severe clinical course of the infection and demonstrates a special ability to form spores [1,2,3,4].

The significant role of the bacterium in the transmission of infections results from its presence in a vegetative form and its ability to form spores. *C. difficile* spores have the ability to survive for a long time on inanimate objects; moreover, they demonstrate high resistance to drying, temperature and many chemical disinfectants, which results in difficulties with the effective decontamination of surfaces and medical equipment and with restriction of the micro-organism’s spread in a hospital environment [5,6]. A significant CDI risk factor is also the patient’s environment [7]. *C. difficile* spores in a hospital environment may survive from a few weeks to over 5 months and may be isolated from a large number of locations in the patient’s environment, e.g., the patient’s bed frame, bed linen, door handles, toilets, sinks, heaters and medical equipment.

The probability of CDI in hospitalised patients who remain in a hospital for more than 2 days and who are undergoing antibiotic therapy is, on average, 27% [8], whereas asymptomatic colonisation of hospitalised patients with *C. difficile* may be observed in the range of 5% to 19% of cases, and its main risk factor is undergoing antibiotic therapy within the last 3 months [9,10]. The clinical picture of *C. difficile* infection can have a very broad range—from mild diarrhoea to symptoms of pseudomembranous colitis, colitis and pathological distension of the colon. CDI is strictly related to weakening of the functioning of the patient’s intestinal microbiome, which most frequently occurs after antibiotic therapy. The remaining factors which facilitate CDI include old age, comorbidities, impaired immunity, nutritional status and sex. The occurrence of CDI is also related to accidental infection with *C. difficile* on hospital wards due to improper hospital hygiene, e.g., through the hands and scrubs of the medical personnel and an ineffective manner of cleaning and disinfecting rooms by the cleaning staff, and thus, insufficient epidemiological and cleanliness control in the hospital [11,12].

### Hospital Environment

The definitions of healthcare-associated infections (HAI) were developed by a team of experts appointed by the European Centre for Disease Prevention and Control (ECDC) in 2009 in order to unify infection diagnostic criteria, including the definition of cases of *C. difficile* infection [11,12,13]. A positive impact on decreased risk of *C. difficile* infections is (should be) provided in a hospital by appropriately trained medical personnel and through the possession of current and implemented cleanliness procedures.

An important procedure which protects against hospital pathogens is proper hand hygiene in medical personnel.

The procedures for the appropriate cleaning and disinfection of hospital rooms are also very important. As part of the maintenance of cleanliness in medical facilities the most frequently used preparations include low-grade non-critical surface-disinfecting agents and washing agents (detergents), such as: ethyl alcohol, sodium hypochlorite, accelerated hydrogen peroxide (AHP), iodophors, phenols, quaternary ammonium compounds (e.g., didecyl dimethyl ammonium bromide and dioctyl dimethyl ammonium bromide), peracetic acid with hydrogen peroxide, glutaraldehyde and sodium dichloroisocyanurate. Unfortunately, some of these agents do not have sporicidal properties, including the destruction of *C. difficile* spores. When eliminating *C. difficile* spores, it is recommended to use sodium hypochlorite, glutaraldehyde or peracetic acid. Additionally ultraviolet radiation (UV-C), steam or hydrogen peroxide aerosol, peracetic acid and ozone are used [14]. All disinfecting agents should be periodically exchanged, in order to ensure that microbes do not become resistant to them.

Correct disinfection of hospital surfaces combined with appropriate inspection may significantly contribute to the elimination and reduction of the number of cases of nosocomial infections [15,16,17]. The goal of the research presented herein was to detect and analyse the degree of contamination of an inanimate internal hospital environment with *C. difficile* spores on three selected wards of a specialist hospital in Poland. The detection of *C. difficile* spores in a hospital environment demonstrates carelessness or failure to follow environmental cleanliness maintenance procedures in a hospital and the potential possibility of the occurrence of other pathogens responsible for nosocomial infections. The tests were conducted using a method of culturing in C diff Banana Broth^TM^ medium.

## 2. Materials and Methods

### 2.1. Description of the Research Unit

Testing of microbiological contamination of the inanimate internal environment of the medical facility for the presence of *C. difficile* spores was conducted once, as a point-prevalence survey, on 13 May 2022 in a specialist hospital in Poland. This hospital has 15 wards, its own outpatient department and diagnostic laboratories. The hospital has a signed contract with the National Health Fund for the treatment of patients from across Poland; it may admit over 450 patients simultaneously, treats over 20,000 patients annually, and has a Certificate of Accreditation for hospital treatment and an ISO 9001 Certificate. It operates in accordance with the regulations in force in Poland, including the Journal of Laws 2022, item 633 on Healthcare Institutions and item 974 on Medical Products [18,19].

In the selected hospital, the main factors for nosocomial infections in years 2019–2020 were *C. difficile*, *Staphylococcus aureus* and *Acinetobacter baumanii*. Even though nosocomial infections caused by *C. difficile* were not first on the list of pathogens in most hospital wards, a decision was made to analyse this pathogen, since in recent years, a rapid increase in the number of infections related to this bacterium has been observed in Poland. *C. difficile* is becoming more and more common among nosocomial infections, not only in Poland but also globally, which means that screening tests for its presence are becoming more and more important [1,2,3,4]. At the same time, tests detecting *C. difficile* with the use of C diff Banana Broth™ are not conducted frequently in Poland.

### 2.2. Characteristics of the Selected Wards

Three hospital wards were selected for the tests: the Internal Medicine Ward, the Intensive Care Unit and the Cardiology Ward. The patients and medical personnel characteristics of these wards for the years 2019 and 2020 are listed in Table 1. 

The selected wards are characterised by increased frequency of nosocomial infections, including ones caused by *C. difficile*. At these three wards the number of nosocomial infections caused by *C. difficile* amounted in 2019 to 69%, and in 2020 to 74% of the infections caused by this bacterium in the entire hospital. For all nosocomial infections from all hospital wards (alarm and non-alarm infections), the Internal Medicine Ward was characterised by *C. difficile* infections at a level of 28% in 2019 and 33% in 2020, the Cardiology Ward at a level of 10% in 2019 and 21% in 2020, and the Intensive Care Unit at a level of 5.6% in 2019 and 4% in 2020. In all hospital wards, infections caused by *C. difficile* in 2019 amounted to 75 cases, whereas in 2020, they amounted to 74 cases. All nosocomial infections included in the list of alarm pathogens in 2019 constituted 218 cases, and in 2020, 237. The Internal Medicine Ward constituted 72 alarm pathogen infection cases (33%) in 2019 and 77 (32%) in 2020; in the Cardiology Ward, 8 (3.6%) cases were noted in 2019 and 7 (3%) in 2020; and in the Intensive Care Unit, 62 (28%) alarm pathogen infection cases were recorded in 2019 and 74 (31%) in 2020.

### 2.3. Microbiological Studies

The collection of samples for the presence of *C. difficile* spores in the analysed wards was conducted in the hours before noon, assisted by an epidemiology nurse and an employee serving as a guide to the medical facility. A hospital environment is contaminated with various micro-organisms, which may be potential reservoirs for the spread of various diseases. However, looking at the goal of the study, the collection of samples was established based on the characteristics of each ward; the number of noted nosocomial infections; the analysis of research conducted in another hospital for the purpose of the PM model described in a study by Kiersnowska et al. (2021), in which critical locations related to the potential presence of *C. difficile* spores were designated; and global literature. The selected sample collection locations included: ventilation and air conditioning grates located in the upper section of patient rooms; windowsills, radiators, sockets and door handles; the frames of patients’ beds; external bed linen of the patients; hand disinfecting agent dispensers located in patients’ rooms and in the corridor; and the handle of the generally accessible kettle. These places are usually accessible to patients and medical personnel and may provide a path of transmission for many pathogens. On the day of the test, cleaning of the rooms in the hospital in the selected wards occurred in the morning hours (before the collection of samples) and immediately as needed. A total of 117 samples (*n* = 117 swabs) were collected from 3 wards, that is: the Internal Medicine Ward (*n* = 40), the Intensive Care Unit (*n* = 37) and the Cardiology Ward (*n* = 40). Using a disposable, sterile flocked swab infused with 0.85% saline solution, the material was collected from inanimate locations in an aseptic manner, and then, the applicators were inserted into vials with Banana Broth medium (C diff Banana Broth™, Hardy Diagnostics, Santa Maria, USA). The vials were locked tightly in order to maintain an anaerobic environment and rapidly transported to the laboratory for incubation. The samples were incubated at 37 °C for 72 h. After incubation, every vial of Banana Broth was controlled for a change in the colouring of the medium. A change in the colour of the medium from red to yellow was considered to be a positive indication for the presence of *Clostridioides* spp. Every vial was tested separately, as a result of which a positive or negative result was determined. The lack of appearance of a yellow colour indicated a negative reaction for the fermentation of mannitol and suggested a lack of *Clostridioides* spp.

C. diff Banana Broth™ (*C. difficile* Brucella Broth with thioglycolic acid and L-cystine or CDBB-TC) is recommended for the culturing and recovery of *C. difficile* spores and vegetative cells from environmental samples. This product is not intended for diagnosis of the disease in humans. C. diff Banana Broth™ is initially reduced, ready for use, very sensitive and has a specific method of detecting *C. difficile*. Since the literature data indicate that other pathogens can also grow on C. diff Banana Broth™ medium, samples with a positive result (change in the colour of the broth to yellow) were subjected to further diagnostics [21,22,23,24]. Positive samples (coloured yellow) were screened in CLO and CDIFF agar media (BioMérieux, Lyon, France) and in Columbia blood agar medium (Graso Biotech, Owidz, Poland). The agar media were incubated at 37°C in anaerobic conditions for 48 h. Afterwards, in order to identify the species of *Clostridioides* strains, a Vitek 2 Compact device (BioMérieux, Lyon, France) was used. In the diagnosed *Clostridioides* strains from the *C. difficile* species, genetic testing was performed using a GeneXpert device (Cepheid GmbH, Krefeld, Germany). Xpert *C. difficile* BT sets were used, which detect the sequence of genes responsible for the production of the B toxin (*tcdB*), the binary toxin (*cdtA*) and *tcdC* base pair deletion at position 117 related to the 027 ribotype strain. All tests were conducted in accordance with the manufacturer’s protocol.

## 3. Results

Among the 117 samples collected from three hospital wards, 18 were positive for the presence of *Clostridioides* spp. (15.4%). *C. difficile* spores detected in six samples (5.1%), *Clostridioides perfringens* in eight samples (6.8%), (1.7%) *Clostridioides sporogenes* in two samples and (0.85%) *Clostridioides innocuum* and *Clostridioides baratii* in one sample each. The tests of gene sequences responsible for the production of toxins among the six *C. difficile* samples demonstrated that four strains produced the B toxin (66.7%) and two strains (33.3%) did not produce toxins. In our study, the binary toxin related to the 027 ribotype was also not detected (Table 2). *C. difficile* spores were detected in the Internal Diseases Ward (*n* = 4, that is, 3.4% of all samples and 10% of samples collected in this ward) on the windowsill next to the window in a hospitalised patient’s room, on a bed frame, on a handle in a patient’s room, and on the handle of a toilet accessible from the corridor; in the Cardiology Ward (*n* = 1, that is, 0.85% of all samples and 2.5% of samples collected from this ward), they were detected on a heater from an external part in the room of a hospitalised patient; and in the Intensive Care Unit (*n* = 1, that is, 0.85% of all samples and 2.7% of samples collected from this ward), they were detected on patients’ bed linens.

*C. perfringens* was identified in the Internal Medicine Ward (*n* = 6, that is, 5.1% of all samples and 15% of samples collected from this ward) and in the Cardiology Ward (*n* = 2, that is, 1.7% of all samples and 5% of samples collected from this ward) in locations such as: handles of the toilet located in the corridor, patients’ bed frames and light switches for the toilet located in the corridor. The remaining environmental species of *Clostridioides*, cultured in four samples were identified in the Cardiology Ward (*Clostridioides sporogenes* (*n* = 1) on the sink tap, *Clostridioides baratii* (*n* = 1) on the light switch in a patient’s room), in the Intensive Care Unit (*Clostridioides innocuum* (*n* = 1) on the ventilation grate) and in the Internal Ward (*Clostridioides sporogenes* (*n* = 1) on the handle of the generally accessible kettle). A graphical interpretation of the obtained results is shown in Figure 1.

## 4. Discussion

Studies demonstrate that there is a problem in hospitals with protecting hospitalised patients against nosocomial infections, and even with deaths caused by these infections, as confirmed by the literature data [25,26,27,28,29]. In total, in all the hospital wards in the studied facility, there were 75 patients infected with *C. difficile* spores in 2019 and 74 patients in 2020, of which death with a *C. difficile* nosocomial infection occurred in 14 (18.6%) hospitalised patients in 2019 and in 15 (20.2%) in 2020. The conducted research confirms the presence of spores in a hospital environment. The results of the tests are confirmed by the hospital’s statistical data. The degree of contamination of the environment with *C. difficile* spores was highest in the Internal Ward, which may partially explain why the largest number of CDI cases in this hospital was present in this ward (56% in 2019, and up to 61% in 2020). In this ward, the average duration of stay for a CDI patient (until discharge from the facility) amounted to 20 days, and the average age of patients was 80 years. The literature data are very similar and confirm that the duration of hospitalisation of patients infected with *C. difficile* was, in most cases, longer than the average stay of the entire studied population of patients. The extended duration of stay due to CDI, among other things, increases of the costs of treatment for patients [30].

In the remaining two wards, the number of patients registered as infected with HAI and the number of nosocomial infections related to *C. difficile* were significantly lower in individual years; this was connected to, among others things, other dominating pathogens (*Acinetobacter baumanii*, *Staphylococcus aureus* and *Escherichia coli*), for which presence in the environment was not analysed in this study. Therefore, in these wards, a lower level of environmental contamination with *C. difficile* spores was noted. It may seem that six positive results for the presence of *C. difficile* spores, including four toxigenic strains, out of 117 samples collected in the selected wards, and four other species of *Clostridioides,* is a relatively low number for confirming contamination of a hospital environment. However, taking into account the locations of *Clostridioides* spp. spore contamination and the fact that only three hospital units were tested, one can establish that the processes and procedures related to the maintenance of cleanliness, the hospital hygiene of medical personnel (including hand hygiene) and general infection risk management are not fully implemented.

In the Intensive Care Unit, patients are prone and at additional risk of pathogen transmission (including of CD spores) from medical personnel during the provision of care and medical procedures. In our study, toxigenic strains of *C. difficile* were detected in the ICU on a patient’s bed linen, which may constitute evidence that they were transferred by medical or auxiliary personnel. In the literature, described cases of CDI in ICUs do not establish the precise location of the occurrence of *C. difficile* spores [29,31,32]. The most frequent location of the occurrence of *C. difficile* spores is the side railing of patients’ beds; however, in our study, they were not detected in this location and were only present on the bed linens [21,33]. Albrecht & Pituch (2013) also point out that a patient’s stay in the Intensive Care Unit itself is one of the reasons for infection with *C. difficile* [34]. Another source of infection in the Internal Medicine Ward and in the Cardiology Ward may also be the patients themselves. A patient walking around with the bacterium colonised on their skin (or in their own bacterial flora) may cause its easy transmission and cross infections to other patients and employees. Patients in these wards frequently do not maintain hand hygiene, they touch various items when moving to support their balance (e.g., walls, railings, the frames of the beds of other patients and medical equipment), and they use the commonly accessible toilets, TV pilots and kettles. This is why appropriately trained cleaning staff working to eliminate micro-organisms from the hospital environment around patients are also very important. In our study, in the Internal Medicine and Cardiology wards, *C. difficile* spores were detected on the surfaces most frequently colonised according to the literature; an exception was the windowsill of the window of hospitalised patients’ rooms [21,33].

In addition to the detection of *C. difficile* in the direct environment of the patient, studies have also confirmed the presence of other pathogenic species, such as *C. perfringens*, *C. sporogenes*, *C. innocuum* and *C. baratii*. These are species which are also detected in other medical facilities and which may constitute a significant disease-causing problem related to HAI [35,36,37,38,39,40,41]. Analysis of the behaviour of hospital personnel was not a direct goal of this study, but it supplements the conducted research analyses. During the study, a medical employee with visibly dirty scrubs which could form a pathogen transmission vector was observed. The literature data indicate that hospital clothing, as well as the hands of the employees, may constitute a significant source and reservoir of disease-causing pathogens. [16,42,43]. A medical employee was also noticed who, without changing their gloves, had direct contact with multiple patients. The literature specifies that contamination occurs through the hands or gloves of personnel. The pathogens most frequently detected on the hands of personnel include: *C. difficile* spores, and VRE (vancomycin-resistant Enterococcus) and MRSA (methicillin-resistant *Staphylococcus aureus*) strains [16,42,44,45]. In a study conducted by Kiersnowska et al. (2022), it was demonstrated that as many as 41% of personnel did not maintain correct hand hygiene procedures in a situation of contact with a patient infected with or suspected of *C. difficile* infection [15]. The presence of *C. difficile* spores in a hospital environment may cause its easy transfer to the remaining patients, but also to medical personnel, who come into contact with other patients and employees.

*C. innocuum* spores were detected on a ventilation grate in the ICU, and according to the literature, heating, ventilation and air conditioning (HVAC) equipment is very susceptible to the accumulation of contaminants and disease-causing pathogens [27,46,47]. *C. innocuum* has natural resistance to vancomycin and may cause *C. difficile*-related diarrhoeal infection [40,41]. Nosocomial infections caused by *C. difficile* spores are an important and complex problem, both epidemiologically and economically, for all hospitals in Poland and worldwide. According to the study by Finn et al. 2021, in the US, the morbidity indicator per 10,000 days of CDI patients amounts to 8.00, whereas in Europe, the highest morbidity indicator per 10,000 days of CDI patients was noted in Poland at a level of 6.18 [48]. In Poland, according to the National Institute of Hygiene, the number of cases of *C. difficile* infections presented in epidemiological reports within the space of the last 9 years (the years 2013–2021) continued to increase [25]. In 2013 a total of 6426 cases were noted, whereas in 2021, as many as 21,174 infections were noted.

There are multiple reasons for the increase in these infections. In the many years of our studies, we have concentrated on the hospital environment surrounding the patient, the hygiene of medical personnel and general hospital hygiene, which are only some of the causes of HAIs. In our studies we would like to demonstrate that these causes have a significant impact on patient safety, and also that incorrect organisation of infection risk management can be encountered in medical facilities in Poland [1,15,26,27,30,42].

## 5. Conclusions

To summarise, the results of our study have confirmed statistical data on medical facilities regarding the incidence of CDI in selected wards. The inanimate environment in the direct vicinity of a patient is a place where *Clostridioides* spores are present. In 117 collected samples, the presence of 18 positive samples of *Clostridioides* spp. was established. Of these, *C. difficile* spores were detected in six samples, *Clostridioides perfringens* in eight samples, *Clostridioides sporogenes* in two samples and *Clostridioides innocuum* and *Clostridioides baratii* in one sample each. Among the six samples of *C. difficile*, a total of four strains which produce the B toxin were cultured, while the binary toxin related to ribotype 027 was not detected. From an epidemiological point of view, the monitoring of pathogen presence is of utmost importance. The research tool used in this study (C. diff Banana Broth^TM^ culturing medium), the assumptions of the PM model (to indicate critical locations) and the analysis of the obtained results indicate the need to undertake more comprehensive and coordinated action to reduce the occurrence of CDI. An increase of knowledge on the risk factors for the presence of infections caused by *C. difficile* may contribute to breaking the CDI chains of transmission.

A hospital environment contaminated with *C. difficile* spores indicates irregularities in multiple aspects of hospital operation. The results of this research should be presented both to medical personnel and to cleaning teams simultaneously to point out the possibility of eliminating the presence of pathogens. The obtained results prove that further work on infections, and on minimising HAI, is necessary, and the procedures and methods used should be systematically checked and expanded with new models supporting the minimisation of infections.

## Figures and Tables

**Figure 1 ijerph-19-15670-f001:**
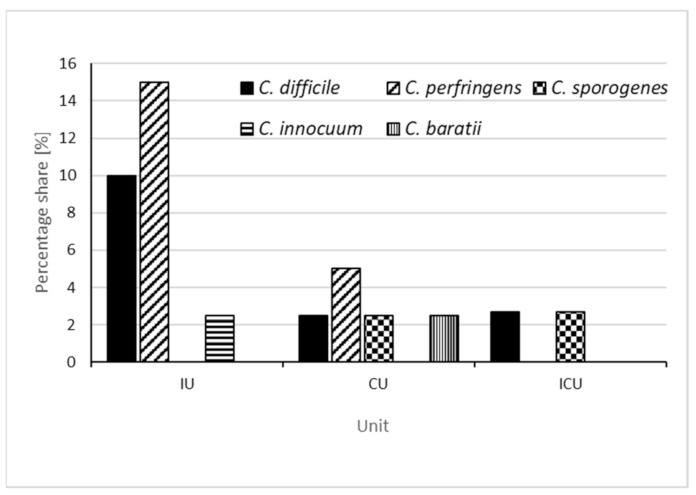
The percentage share of *C. difficile*, *C. perfringens*, *C. sporogenes*, *C. innocuum* and *C. baratii* detected in individual wards, calculated as a ratio of the number of samples collected in a given ward (IU—internal unit, CU—cardiology unit, ICU—intensive care unit).

**Table 1 ijerph-19-15670-t001:** Characteristics of the selected wards.

Analysed Parameters	Internal Medicine Ward		Intensive Care Unit		Cardiology Ward	
	2019	2020	2019	2020	2019	2020
The number of patients registered as infected with HAI, infection outside of the hospital and colonisation (related to the infection in general)	1560	1087	326	380	115	90
The number of nosocomial infections (in general—alarm factors ^1^ and non-alarm factors)	151	136	107	126	39	24
Number of nosocomial infections with *C. difficile*	42	45	6	5	4	5
Dominating factor of nosocomial infections in a given ward	*C. difficile*	*C. difficile*	*Acinetobacter baumanii* (*n* = 25)	*Acinetobacter baumanii* (*n* = 25) and *Klebsiella pneumoniae* (*n* = 20)	*Staphylococcus aureus* (*n* = 6)	*C. difficile* and *Escherichia coli* (*n* = 5)
Number of deaths of patients infected with *C. difficile*	5	8	2	2	2	1
Number of deaths of infected patients (in general)	33	30	27	33	8	1
Number of women of infected with *C. difficile*	26	18	2	2	0	2
Number of men infected with *C. difficile*	16	27	4	3	4	3
Number of doctors	62	73	20	27	20	24

^1^ alarming agents—biological pathogen of particular virulence or resistance, life-threatening [20].

**Table 2 ijerph-19-15670-t002:** The results of *Clostridioides* spp. diagnostics and the types of produced toxins.

Species	Number of Positive Samples	Internal Ward	Intensive Care Unit	Cardiology Ward
*C. difficile*	6	2—B toxin 2—no toxins	1—B toxin	1—B toxin
*C. perfringens*	8	6		2
*C. sporogenes*	2	1		1
*C. innocuum*	1		1	
*C. baratii*	1			1

## Data Availability

The data presented in this study are available on request from the corresponding author.

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
