# Peer review of "Detection and Analysis of Clostridioides difficile Spores in a Hospital Environment"

_ijerph, 2022, doi:10.3390/ijerph192315670_

Round 1
Reviewer 1 Report
I appreciate the efforts and dedication about what authors have studied. Authors conducted this study to analyze the degree of contamination of hospital environment with Clostridioides difficile spores.
1. The results of this study are light. However, the introduction and discussion are too abundant. Authors could cut down some well-known information of Clostridioides difficile in introduction.
2. The degree of contamination of hospital environment with Clostridioides difficile spores seem to be difficult statistic analyzed from small samples sizes. Clostridioides difficile spores were just detected in six samples. Could authors present the different degree of contamination between these three selected wards? If the differences exist between these three selected wards, could authors indicate the factors which are related to the differences, such as patient condition or compliance of cleanliness control?
3. Clostridioides perfringers in eight samples, Clostridioides sporogenes in two samples and one sample each of Clostridioides innocuum and Clostridioides baratii were detected. Were these bacteria true pathogens or just colonization? Were any patients infected by these pathogens in this study? And what is the impact of these isolates to the infection control?
Thank you.
Author Response
Reviewer 1
I am very grateful for the presented critical remarks to the manuscript. We have tried to take into account the presented suggestions.
1.The introduction and discussion have been improved. A detailed analysis for the detection of C. difficile was added according to suggestions from other reviewers.
- Differences between departments are given, at the same time data on infections are presented in the table hospitals in these departments and cases of C. difficile.
- In this study, patients present at the hospital were not diagnosed for infections bacteria of the genus Clostridioides. Hospital statistics on the number were used cases of C. difficile. The remaining detected microflora, it is not known what its origin is, however, there is information in the literature that it may also be potentially pathogenic.

Reviewer 2 Report
Dear editor and authors:
The work seems to me a very professional performance as nowadays the identification of bacterial species is necessary because of the problem of species-dependent drug resistance. The work is good but it must improve several things in order to be published.
1. We must recognize that in order to be able to say that it is C. difficile it is necessary to perform more specific tests for its identification, such as the use of Molecular Biology (PCR).
2. The test used in this case is phenotypic, therefore genotypic identification is necessary.
3. It is important to mention that when using these disinfectants, they should be changed periodically so that the bacteria do not become resistant to them. If two disinfectants are going to be used that can destroy the spore, the change should be made every 15 days and repeated the following month.
4. Is would be interesting to mention the correct way to wash hands.
5. The work is not original as other works have been done using this culture medium in Poland: KabaÅ‚a M, Gofron Z, Aptekorz M, Burdynowski K, Harmanus C, Kuijper E, Martirosian G. Detection of Clostridioides difficile in hospital environment by using C diff Banana Broth™. Anaerobe. 2022 Feb;73:102408. doi: 10.1016/j.anaerobe.2021.102408.
6. The closest relative of Clostridium difficile is Clostridium mangenotii with a 94.7% similarity value, making it necessary to use Molecular Biology.
Author Response
I am very grateful for the presented critical remarks to the manuscript. We have tried to take into account the presented suggestions. The assumption of the study was to detect the spores in an inanimate environment of the hospital wards, which is why critical points selected previously based on the PM model were subjected to the analysis. Concerning the selection of locations for the sampling, which based on the PM model were considered to be critical points, and on the basis of the spore detection method (C diff Banana BrothTM), which is a medium more frequently used in other countries, and rarely in Poland, we consider this to be an original study for the selected facility in Poland. The Introduction and Research methods were corrected. The issue of disinfecting agents was pointed out and literature on hand hygiene was quoted. The previous manuscript did not specify full research methods, since the goal of the study was to detect and to assess the frequency of occurrence of C. difficile spores, using the C. diff Banana Broth™ medium in the inanimate environment near the patient, and not the precise diagnostics of clinical samples collected from patients with CDI. Since studies for the detection of sequences of genes responsible for the production of toxins were conducted, the description of methodology was expanded, and the results of the studies were quoted in the interpretation of the results. The results of the studies were described in more detail, taking into account the toxigenic strains. The summary was also expanded. Once again, I am very grateful for the presented critical remarks to the manuscript.

Reviewer 3 Report
Authors present a study that seeks to analyse the degree of contamination of hospital environment with Clostridioides difficile spores. The culture they used was performed using a C diff Banana Broth medium, which enables the germination of the spores of these bacteria. Their samples were collected from inanimate objects within the hospital. The results shown the occurrence of 18 positive samples of Clostridioides spp., out of which Clostridioides difficile spores were detected in 6 samples, Clostridioides perfringers in 8 samples, Clostridioides sporogenes in 2 samples and 1 sample each of Clostridioides innocuum and Clostridioides baratii. Their studies have demonstrated the importance of continuous supervision of nosocomial infections, in particular when a risk group is exposed.
The method seems interesting, and the results are promising, but perhaps it would be more interesting if a more exhaustive statistical study of the results obtained had been carried out. The authors only show the specific case of a study performed for n patients, so it would be necessary to extrapolate to a larger population with a slightly more in-depth statistical study. They only apply percentages of occurrence of one type of case or another.
Also, there are some sentences and typos that should be rewritten:
- Lines 27 to 31: Rewrite this paragraph. The verb “become” is abused. Change it.
- Lines 122,123,124,126 should begin with a capital letter.
- In line 235, change “37oC” into “37º”.
To conclude, there are a lot of self-references (almost 10 per author). Maybe a lot. Remove the less relevants.
In conclusion, I would accept the article if authors include a more exhaustive statistical study of the subject, not just focusing on giving mere percentages.
Author Response
I am very grateful for the presented critical remarks to the manuscript. We have tried to take into account the presented suggestions. The text was subjected to editorial corrections. Since the manuscript concerns the continuation of studies of C. difficile, and our earlier works concerned issues related to CDI, the new PM model, the costs of treatment of patients and hospital personnel hygiene, which is why self-citations were introduced in the manuscript out of necessity. We have tried to reduce them, but it was not always possible.

Reviewer 4 Report
Summary
In the current study, Zofia and colleagues identified Clostridioides difficile and related species infections in the Specialist hospital environment in Poland. The authors have discussed and cited the literature and current studies associated with C. difficile infections in the hospital environment. Such ongoing studies have a more significant impact on real-time problems.
Comments
1. First two lanes of the abstract are poorly written; I would advise the authors to re-write the primary standpoint of this study
2. Considering the importance of the study in deciding nosocomial infections, the sample size of the study is significantly less
3. Authors should consider the including positive results obtained from the CLO and CDIFF agar Media
4. The authors are solely dependent upon the data from Vitek 2 Compact device and have drawn significant conclusions from the study. The data should be included in the results and discussion
5. Authors should also consider verification of Clostridioides stains using DNA sequencing (16S ribosomal RNA gene is the most widely used marker gene in microbial ecology)
I believe it is a very nice and informative study with valuable data. Still, the way of representation is inferior and needs to work on, including figures, data representation, and English.
Author Response
I am very grateful for the presented critical remarks to the manuscript. We have tried to take into account the presented suggestions. The number of samples collected was explained. On the average, 37-40 samples were collected, which I consider representative for every analysed ward. Even though the Internal Medicine Ward is the largest, more samples were not collected there, since an assumption was made that we collect a similar number of samples on every ward. The assumption for the study was to detect spores in the inanimate environment of the hospital, and not the diagnosis of patients for CDI. This is why the analysis of critical points established previously based on the PM model was conducted. The Introduction and Research methods were corrected. The previous manuscript did not specify full research methods, since the goal of the study was to detect and to assess the frequency of occurrence of C. difficile spores, using the C. diff Banana Broth™ medium in the inanimate environment near the patient, and not the precise diagnostics of clinical samples collected from patients with CDI. Since studies for the detection of sequences of genes responsible for the production of toxins were conducted, the description of methodology was expanded, and the results of the studies were quoted in the interpretation of the results. The results of the studies were described in more detail, taking int account the toxigenic strains.

Reviewer 5 Report
In their paper Kiersnowska et al show the result of a prospective study performed culturing swabs of inanimated objects in a hospital in Poland with the aim of quantifying the Clostridioides spp spores’ contamination.
The article in my opinion needs to be re-written with a huge English review, some sentences are difficult to understand. I strongly suggest a native speaker review.
Moreover, I would suggest to write in a less narrative way.
In almost the whole paper C. perfringensis misspelled
In particular
Abstract
Page 1 line 10: CDI is not THE most frequent HAI, but one of the most frequent
Introduction
Introduction is way too long. I would not talk about the clinical features of CDI as it is not the aim of the paper.
Page 1 lines 32-43: is not very clear. It seems a collection of personal opinions from the authors.
Methods
The Methods section should be divided into paragraphs (e.g., microbiology, ward choice ecc).
If it is, as I assume, a prospective study, it should be stated.
Table 1 is not necessary, at least not in this form. Please consider simplifying it.
The last sentence (page 6 lines 237-240) is not a method, but a personal view of the authors.
Results
It would have been interesting to know if a higher degree of contamination was associated to a higher incidence of CDI (Internal unit vs cardiology and ICU)
Discussion
Too long and narrative
Author Response
I am very grateful for the presented critical remarks to the manuscript. We have tried to take into account the presented suggestions. The Introduction and Research methods chapters were corrected. The previous manuscript did not specify full research methods, since the goal of the study was to detect and to assess the frequency of occurrence of C. difficile spores, using the C. diff Banana Broth™ medium in the inanimate environment near the patient, and not the precise diagnostics of clinical samples collected from patients with CDI. Since studies for the detection of sequences of genes responsible for the production of toxins were conducted, the description of methodology was expanded, and the results of the studies were quoted in the interpretation of the results. The criteria for the selection of wards for the study were described in more detail, the table with the characteristics of the wards and of the hospital was improved. In the Study results the comparison of the wards was described, pointing out the number of recorded cases of C. difficile. The discussion of the results and the summary was improved.

Round 2
Reviewer 2 Report
The authors corrected the article, so I suggest that it be published in the format it is in. Regards
Author Response
Bardzo dziękuję za przyjęcie dostarczonych poprawek i przyjęcie manuskryptu.
Reviewer 4 Report
The authors have improvised every section of the manuscript and I have no further comments
Author Response
Thank you very much for accepting the provided amendments and for accepting the manuscript.
Reviewer 5 Report
I thank the authors for the review and the reply.
I still believe it needs a native speaker editing.
As an example, there is no need to write “bacterium” after C. difficile.
Introduction
Still too long: lines 27 to 36 are not necessary.
Lines 45-46 the 027 ribotype it is not called this way in Europe, but it is more common in Europe
Lines 62-64: “A 70-85% of patients are infected with C. difficile during hospitalisation. An asymptomatic colonisation with C. difficile in the hospitalised patients may be observed in ap- 63 prox. 25% of patients.” References are missing.
Lines 63-64: approximately should be written in the correct way, no abbreviation should be used.
What Is the Hospital environment section? If it as, as I assume, part of the introduction, then it is way too long.
The type of study is still missing (one-day point-prevalence study).
Too narrative and wordy as the discussion.
It should be simplier and focused of the findings: a higher prevalence of contamination was found in the internal medicine ward and this reflects the higher number of CDI.
A lot is said about cleaning procedures, but what about contact precautions?
Author Response
I am very grateful for the presented critical remarks to the manuscript. We have tried to take into account the presented suggestions. The text was subjected to editorial corrections. The Introduction and Discussion sections were shortened and improved. The type of the conducted survey was specified. The Hospital Environment section was shortened and improved - the Hospital Environment section reflects the conditions in the hospital and the potential location for the presence of pathogens, including CD spores. It describes the most frequent methods of transferring of CD spores and the methods for their elimination. The methods for maintaining cleanliness at a medical facility were indicated in accordance with suggestions made by other Reviewers. We have accepted your suggestions by shortening this section to a necessary minimum. Literature was supplemented. Concerning contact precautions of the medical personnel at the Intensive Care Unit, the potential role of medical personnel in the transmission of CD spores during care and medical procedures performed on prone patients was pointed out. Similarly, for the Internal and Cardiology Wards we stipulated that cross-contamination may occur between moving patients and personnel.
The study did not analyse the cases of CD in hospitalised patients. The number values provided of the number of CD infections at individual wards and at the entire medical facility are only specified as background for explaining hospital infections that may occur.